# Modulating Growth Factor Receptor Signaling to Promote Corneal Epithelial Homeostasis

**DOI:** 10.3390/cells12232730

**Published:** 2023-11-29

**Authors:** Kate E. Tarvestad-Laise, Brian P. Ceresa

**Affiliations:** 1Department of Pharmacology and Toxicology, University of Louisville, Louisville, KY 40202, USA; 2Department of Ophthalmology and Vision Sciences, University of Louisville, Louisville, KY 40202, USA

**Keywords:** cornea, corneal epithelium, growth factor, growth factor receptor, c-Met, wound healing, corneal nerves

## Abstract

The corneal epithelium is the first anatomical barrier between the environment and the cornea; it is critical for proper light refraction onto the retina and prevents pathogens (e.g., bacteria, viruses) from entering the immune-privileged eye. Trauma to the highly innervated corneal epithelium is extremely painful and if not resolved quickly or properly, can lead to infection and ultimately blindness. The healthy eye produces its own growth factors and is continuously bathed in tear fluid that contains these proteins and other nutrients to maintain the rapid turnover and homeostasis of the ocular surface. In this article, we review the roles of growth factors in corneal epithelial homeostasis and regeneration and some of the limitations to their use therapeutically.

## 1. Introduction

The corneal epithelium is the first anatomical barrier between the environment and the eye. Its structure is critical for proper light refraction onto the retina. The epithelial layer prevents pathogens (e.g., bacteria, viruses) from entering the immune-privileged eye and causing inflammation and scattering of light [1]. The corneal epithelium is the most densely innervated tissue in the body [2], so trauma is exceptionally painful. The healthy eye is continuously bathed in tear fluid containing proteins that are necessary for the rapid turnover and maintenance of the ocular surface [3]. Wounds to the epithelium promote the upregulation of growth factors in the tear fluid [4], exhibiting their importance in corneal wound healing. If wounds to the epithelium are not resolved quickly or properly, they can lead to infection, fibrosis, and ultimately blindness. In this article, we review the roles of growth factors in corneal epithelial homeostasis and regeneration as potential therapies and discuss some limitations to their use.

## 2. Corneal Epithelium

### 2.1. Anatomical Structure

The human cornea is comprised of 3 cellular layers separated by two distinct collagenous interfaces (Figure 1). The epithelium lies most anterior in the tissue, separated from the stroma by Bowman’s Layer. The stroma lies anterior to Descemet’s membrane, which separates it from the endothelium, the most posterior cellular layer. The endothelium interfaces with the aqueous humor and allows nutrients to diffuse to other corneal cells. It also regulates the hydration of the stroma, which is crucial for visual clarity [5].

The epithelium is made of 5–7 layers of epithelial cells [1]. Corneal epithelial cells arise from limbal stem cells, which migrate to the central cornea and differentiate into basal cells [6]. As basal cells age, they differentiate and migrate anteriorly to the surface, where eventually they will reach their squamous form and shed [6]. This is a very rapid process, with the turnover of the corneal surface occurring every ~10 days [7,8]. This is advantageous for the restoration of the epithelial surface if it is ever damaged by trauma, surgery, disease, or drug side effects. The current model of corneal epithelial homeostasis is the XYZ hypothesis, where movement of limbal stem cells towards the central cornea (X), plus basal cell proliferation and differentiation in a vertical direction (Y), equals the loss of superficial squamous cells from the epithelial surface (Z) [6,9].

### 2.2. Tear Fluid

Because the cornea must remain transparent, there are no blood vessels to bring nutrients to the cells that reside there; rather, epithelial cells rely on the tear fluid and the aqueous humor to bathe and supply oxygen [10,11]. Epithelial cells facilitate uptake of nutrients via microvilli on its superficial layer, which allow for greater surface area interaction between the cornea and tear fluid [12]. Along with supplying the necessary molecules to maintain corneal homeostasis, the tear fluid also serves as the first protective layer of the ocular surface. It forms the barrier between the epithelium and the external environment.

In healthy eyes, the tear fluid consists of three phases. The most anterior phase is the lipid phase, or meibum, which originates in the meibomian glands located in the upper and lower eyelids. The meibum consists of different lipids and mucins that function to ensure an even spread of tear film over the surface of the eye and prevent evaporation of the aqueous layer [13]. The middle aqueous phase is created in and secreted from the lacrimal glands, which reside in the anterior lateral orbit above each eye. The aqueous layer contains mucins, electrolytes, antioxidants, and protein-tlike growth factors that contribute to the homeostasis of the ocular surface [12]. Lastly, the innermost mucin layer is secreted by the goblet cells of the conjunctiva [1,4,14,15,16].

### 2.3. Corneal Nerves

Between the epithelial cell layers is an intricate network of neurons that make the cornea the most densely innervated tissue in the body, with 7000 nociceptors per mm^2^ [2]. Corneal nerves vary in function, but the majority are sensory nerves that respond to touch, changes in temperature, and pain [17]. There are other types of nerves present in the epithelium as well, like peripheral nerves that branch from the superior cervical ganglion and supply sympathetic innervation [18,19,20]. Evidence of parasympathetic innervation in humans from the ciliary ganglion is limited, though it has been found in some animal models [2,19].

Most corneal epithelial nerves branch from the trigeminal nerve and enter the cornea radially at the stromal level. Intraepithelial corneal nerve endings (ICNs) branch upwards from the subbasal plexus, which resides beneath the epithelium but above the stroma (Figure 1) [2]. Many synapse through the lacrimal nucleus in the pons and connect to the facial nerve [21], which in turn activates the meibomian and lacrimal glands to promote production and secretion of tears [22,23]. Other outcomes of ICN activation include blinking to remove foreign objects, watering of the eye, and wound healing [2]. The increase in the production of growth factors after stimulation contributes greatly to the healing response of the cornea.

The ICNs are intimately connected with the corneal epithelial cells and are critical for their health. Because the cornea is transparent for proper light passage, ICNs shed their myelin sheaths and use the basal epithelial cells as their Schwann cell surrogates [24] (though new techniques have discovered evidence of non-myelinating Schwann cells present in the cornea [25,26,27]). Further, when primary corneal epithelial cells and trigeminal neurons are co-cultured, neurite outgrowth is increased [28]. Both cell types release factors that act on the other to help maintain homeostasis of the corneal surface. Nerve health directly impacts the health of the ocular surface.

## 3. Growth Factors and Their Receptors

The maintenance of a healthy cornea is driven by growth factor receptor signaling. These soluble proteins bind to their cognate receptor to induce biochemical changes inside the cell to alter its biology. For growth factors to exert their activity, their cognate receptor must be expressed on their target cells at a sufficient density to activate intracellular effectors. Although growth factors have biological roles in embryogenesis, organogenesis, and angiogenesis (and more), in mature and healthy ocular tissues they primarily contribute to maintaining homeostasis [29,30,31].

### 3.1. Receptor Expression

Many families of growth factors and their receptors are expressed in the corneal epithelium, including the platelet-derived growth factor receptor family (PDGFR), vascular endothelial growth receptor factor family (VEGFR), epidermal growth receptor factor family (EGFR), fibroblast growth factor receptor family (FGFR), insulin-like growth factor receptor family (IGFR), and hepatocyte growth factor receptor family (HGFR, or c-Met) (Table 1). When stimulated in the corneal epithelium, these receptors promote cell proliferation, migration, and differentiation to aid in corneal re-epithelialization.

### 3.2. Growth Factor Expression

The growth factors that stimulate these receptors can originate from the tear fluid, the corneal nerves, and the epithelial cells themselves. They can also be synthesized in fibroblast-like cells (i.e., keratocytes), which secrete them to act in a paracrine manner on the epithelial cells nearby (Figure 2). Many growth factors are expressed in basal tears, but their levels are dynamic based on the stresses to the eye.

In mice, the wounding of the cornea increases growth factor presence in the aqueous phase of tear fluid. This delivers a higher concentration to the epithelium, promoting cellular proliferation, migration, and differentiation to promote re-epithelialization [32,33,34]. For example, human tear hepatocyte growth factor (HGF) mRNA levels increase ~1400% from the pre- to post- photorefractive or phototherapeutic keratectomy state [32]. The release of growth and neurotrophic factors in the tear fluid in vivo are, in part, controlled by the corneal nerves [35,36]. Nerves release neurotrophic factors/neuromodulators [i.e., substance P, calcitonin gene-related peptide (CGRP), acetylcholine, and vasoactive intestinal peptide (VIP)] that maintain the ocular surface [36]. Substance P and CGRP specifically have been shown to aid in corneal epithelial homeostasis and turnover [36]. 

Endogenous growth factors can be synthesized in a pro-form and associate with the cell membrane until they are processed by matrix metalloproteinases (MMPs) or a disintegrin and metalloproteases (ADAMs) [37,38]. The released soluble form is then able to bind and activate its cognate receptor. In in vitro corneal epithelial cell scratch wounds, MMP activity is increased, allowing for greater HB-EGF (heparin-binding EGF) processing and cleavage, resulting in the activation of the EGFR [39,40,41,42].

**Figure 2 cells-12-02730-f002:**
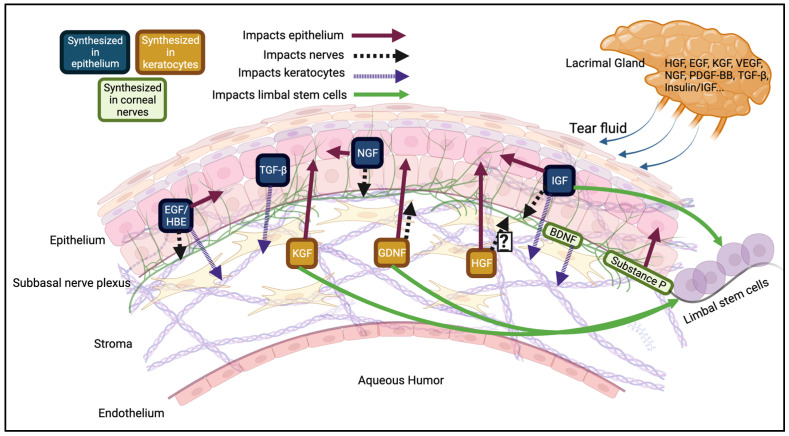
Growth factor expression in the corneal epithelium. A sample of some of the main mediators of corneal epithelial homeostasis. Many growth factors, like EGF, HB-EGF, NGF, and TGF-β are synthesized in the epithelium and act in an autocrine manner, where they bind to epithelial cells, or a paracrine manner, where they bind to nearby cells, like keratocytes or corneal nerves. Alternatively, proteins like HGF and KGF are expressed in keratocytes and diffuse anteriorly to the corneal nerves and epithelium. Some proteins synthesized in the cornea also act on the adjacent limbal stem cells to promote their proliferation. Corneal nerves release neurotrophic factors like substance P and brain derived neurotrophic factor (BDNF) that act in autocrine or paracrine manners [36]. Many of these growth factors can be synthesized in multiple layers of the cornea. Finally, growth factors can reach the corneal epithelium via the lacrimal gland and the tear fluid or through the aqueous humor (regulated by the endothelium). A more comprehensive list of growth factors is in Table 1. Created with BioRender.com (accessed on 17 November 2023).

### 3.3. Receptor Mechanism of Action

Growth factor receptors have comparable mechanisms of action across the different families. Most growth factor receptors are embedded in the cell membrane in a monomeric form (others, like the insulin receptor, are already present in a dimerized form [43]).

**Table 1 cells-12-02730-t001:** Growth factor receptor biology and negative regulators in the corneal epithelium.

Receptor	Factor	Effects on Corneal Wound Healing	Regulation by CBL
c-Met	Hepatocyte Growth Factor (HGF)	Enhances re-epithelialization rates [44] Promotes epithelial cell migration and proliferation [45,46] Suppresses inflammatory signaling mediators [44,47] Promotes synaptogenesis [48], sufficient for peripheral nerve outgrowth [49,50,51] Involved in angiogenesis [52,53,54,55]	c-Cbl [45,56,57] Cbl-b [45,57]
EGFR	Epidermal Growth Factor (EGF)	Enhances re-epithelialization rates [58,59] Promotes epithelial cell migration and proliferation [60,61,62,63,64]	c-Cbl [58,65,66] Cbl-b [66] Cbl-c [67]
	Heparin binding-EGF (HB-EGF)	Enhances re-epithelization rates [68,69]	
	Betacellulin (BTC)	Enhances re-epithelialization rates [68] Promotes limbal stem cell proliferation [70]	
	Transforming Growth Factor-α (TGF-α)	Enhances re-epithelialization rates [68] Promotes EGFR recycling, enhancing wound healing [71]	
KGFR	Keratinocyte Growth Factor (KGF)	Enhances re-epithelialization rates [31,72,73] Promotes limbal stem cell proliferation [74]	c-Cbl [75]
FGFR	Fibroblast Growth Factor (FGF)	Enhances re-epithelialization rates [76] Promotes differentiation of keratocytes to fibroblasts [77,78] Promotes stromal fibroblast proliferation [79] Can induce corneal neovascularization [80]	c-Cbl [81]
IGF-1R; IGF-2R	Insulin Growth Factor (IGF) Insulin	Promotes epithelial cell migration and proliferation [82] Promotes limbal stem cell differentiation [83] Synergistic effects with substance P to enhance wound closure [84] Regulates keratocyte organization network [82] Insulin enhances healing in CE cells by transactivation of EGFR [85,86]	c-Cbl [87] Cbl-b [88]
PDGF-αR & PDGF-βR	Platelet Derived Growth Factor (PDGF)	Promotes migration of keratocytes [31] Enhances epithelial cell migration in presence of fibronectin [31,89] Enhances endothelial cell proliferation [31]	c-Cbl [90] Cbl-b [90]
VEGFR 2	Vascular Endothelial Growth Factor (VEGF)	Enhances re-epithelialization [91] In combination with IL-17, necessary for efficient corneal nerve regeneration [92] Can induce corneal neovascularization [80]	c-Cbl [93]
TrkA	Nerve Growth Factor (NGF)	Enhances re-epithelialization rates [94,95] Promotes epithelial cell migration and proliferation [94] Improves nerve density [96]	c-Cbl [97] Cbl-b [98]
TrkB	Brain derived neurotrophic factor (BDNF)	Stimulates proliferation of keratocytes, but not epithelial cells [99]	c-Cbl [100,101]
TGF-βR	Transforming Growth Factor-β (TGF-β_1,2,3_)	Isoforms 1 and 2: Inhibits corneal epithelial cell proliferation, stimulates keratocytes [102,103] Antagonize EGF, HGF, and KGF-induced corneal epithelial cell proliferation [104,105] Promotes scar formation in stroma [77,106,107] Isoform 3: Enhances corneal wound healing, does not have fibrotic effects [108]	c-Cbl [109] Cbl-b [109,110]
RET/GFR-α	Glial cell line- derived neurotrophic factor (GDNF)	Enhances re-epithelialization rates [111,112] Promotes neurite outgrowth and maintains density of nerves [111] Suppresses inflammatory cytokine signaling, aids in limbal stem cell survival [113]	c-Cbl [114] Cbl-c [115]

Upon ligand binding, they undergo a conformational change and dimerize, and the intracellular kinase domains are brought close enough together to auto/trans-phosphorylate each other. Some receptors have ligand-stimulated tyrosine kinase activity [receptor tyrosine kinases (RTKs), i.e., c-Met and EGFR] and others have ligand-stimulated serine-threonine kinase activity [i.e., transforming growth factor-β (TGF-β) receptor]. Receptor phosphorylation is the unifying feature that allows for the docking and activation of downstream effectors [i.e., mitogen activated protein kinase (MAPK), phosphoinositide 3-kinase (PI3-K), signal transducer and activator of transcription 3 (STAT3), etc.]. Effectors translate the binding of extracellular ligands into intracellular biochemical signaling. The cell response is determined by how long individual effectors are active [116]. The contributions from each effector is governed by the spatial and temporal regulation of the receptor [116].

#### Alternative Signaling Mechanisms

Growth factor receptors interact with many cell surface molecules including mucins, plexins, integrins, other receptor tyrosine kinases (RTKs), and transient receptor potential (TRP) channels. Of these, TRP channels are particularly important regulators of the corneal epithelium, partly due to their expression in intraepithelial corneal nerve endings. In addition, TRP channels are present on the epithelial surface [117]. While they have their own roles in corneal homeostasis, the activation of these channels can lead to the stimulation of growth factor receptors: the treatment of human corneal epithelial cells (HCECs) with capsaicin, a potent agonist of the TRP vanilloid 1 (TRPV1) channel, induces the shedding of HB-EGF and activation of the EGFR [118,119]. There is evidence in other cell types (liver cancer cell lines, human prostate cancer lines, renal tubular cell lines) in which treatment with HGF increases the expression and activity of TRPC6, which contributes to cell proliferation [120]. Further, TRPV1′s Ca^2+^ influx channel activity is necessary for HGF-induced migration in HepG2 cells [121], so TRPV1 activation may be required in corneal epithelial cell motility as well.

### 3.4. Growth Factor Alterations in Disease States

The importance of growth factors can be seen by their varying levels in pathological conditions. Tear dysfunction—pathologies of tear fluid composition and make-up—is a leading cause of corneal epithelial disease and irritation [122]. The 2017 Dry Eye Workshop II (DEWS II) within the Tear Film and Ocular Surface society (TFOS) defined different types of Dry Eye Disease (DED). Within each of these subcategories (i.e., Sjögrens syndrome, aqueous deficient dry eye, meibomian gland dysfunction) the EGF concentration in the tears differs [123,124]. It remains to be determined whether changes in EGF levels are a cause or an effect of DED. DED also presents with upregulated MMPs [125,126], reflecting the eye’s need for more growth factor. The MMP inhibitor GM 6001 prevents wound-dependent EGFR activation, presenting as a delay in healing time, consistent with a block in the processing of HB-EGF [127].

Many disorders, including neurotrophic keratitis (NK), DED, and diabetes present with decreased corneal innervation. Less sensitivity to external stimuli and/or decreased connectivity with the lacrimal gland manifests as lowered tear production and secretion. This in turn decreases growth factor delivery to the cornea and manifests as delayed corneal wound healing [36,128,129,130]. Neuromodulators like substance P, CGRP, and NGF are downregulated in DED [131]. However, research is still necessary to determine if the lack of growth factor is causing the pathology or if the disease is causing the lack of growth factor.

There is ample literature discussing the changes in growth factors [132] in other corneal disorders like keratoconus [133], diabetes [133,134], bullous keratopathy [133], ocular rosacea [135], inflammatory surface diseases [136], and pseudomonas aeruginosa keratitis [137,138]. There are also genetic dimorphisms that impact corneal health. Many corneal dystrophies are manifested through single nucleotide polymorphisms (SNPs) in growth factor genes like TGF-β [139] and HGF [140] (though this has only been seen in certain demographics and not others [141,142]).

## 4. Growth Factor-Mediated Corneal Epithelial Restoration

### 4.1. The Role of Growth Factors for Corneal Epithelial Homeostasis and Restoration

The first evidence of growth factors playing a role in ocular biology came with the discovery of epidermal growth factor (EGF) in 1962 by Stanley Cohen [143]. His seminal discovery showed that addition of EGF accelerated the eyelid opening in newborn mice. Thirty years later, work by Zieske et al. demonstrated that introduction of EGF to debrided mouse corneas accelerated the rate of wound healing [144].

Other early studies demonstrated that FGF1 mRNA is upregulated 6-48h following corneal epithelial burning with a steady return to basal over 6 days [145]. This transient increase in ligand coupled with the presence of receptors on the cell surface point towards the use of exogenous growth factors as a natural way to enhance restoration of the epithelial surface.

### 4.2. Opportunities for the Use of Growth Factors

Due to the rapid turnover of the corneal epithelium, superficial scratches heal within 24–72 h in healthy individuals [146]. However, individuals who have diseases like diabetes [147,148,149] or are taking RTK inhibitors as anti-cancer therapy [150,151,152] often present with recurrent corneal erosions, leading to discomfort and potential loss of vision. Additionally, those who undergo corneal transplants or LASIK surgeries stand to benefit from pharmacologic agents that could help accelerate the healing process, alleviate pain associated with damaged corneal epithelium, aid in nerve regeneration, and prevent possible infection. Such agents also have the potential to help individuals undergoing limbal stem cell transplants by accelerating the restoration of the epithelial layer. Finally, compounds that accelerate the EGFR-mediated responses that promote corneal epithelial homeostasis will further our understanding of corneal epithelial biology as well as help in the development of the epithelial layers of artificial, bioengineered corneas.

#### 4.2.1. Current FDA-Approved Growth Factor Therapies: Amniotic Membrane

Currently, there are limited options available to accelerate corneal epithelial wound healing. Amniotic membrane (AM) has a long history in tissue restoration including in regeneration of skin [153,154,155], in the dental clinic [156], as well as in ocular surface healing [157]. AM also has potent anti-inflammatory effects [158,159] and is used to prevent infection following surgery or burns [160,161]. AM is derived from placental tissue and consists of a single layer of epithelial cells, a basement membrane, an avascular matrix of connective tissue, and many pro-regenerative biomolecules, including EGF, HGF, FGF, and multiple cytokines [162,163].

The first use of AM on the ocular surface Is credited to de Rötth in 1940 [164], and since then, it has been used extensively in research and in the clinic. However, AM is limited in comfortability for the patient, how it is preserved, the variability of sources that it comes from, the types of corneal wounds that it can aid, and it does not address the underlying pathology [165,166,167,168]. However, it is still used in the clinic and is efficacious for healing some wounds [169], though the effects of the growth factors contained in AM have not been well-tested. Another promising option that has yet to be FDA-approved is the use of hydro- or collagenous gels that slowly release growth factors over time [170,171], much like AM. Further research is necessary to optimize these products for broader clinical use.

#### 4.2.2. Current FDA-Approved Growth Factor Therapies: Recombinant Growth Factors

Corneal epithelial wounding insults both the epithelial cells and the corneal nerves. There is ample evidence that corneal nerves can regenerate after injury [111,128,172,173]. Part of this ability is due to epithelial cells and keratocytes releasing growth factors that act on corneal nerves including EGF, NGF, BDNF, and GDNF [99]. Topical treatment with growth factors like NGF and vascular endothelial growth factor (VEGF) can aid in corneal nerve regeneration and corneal healing [91,173]. Treatment to mouse corneal wounds with a dopamine receptor (D1 and D2) agonist increased the levels of NGF present in the cornea and promoted both nerve regeneration and re-epithelialization [174]. Further, combination therapy of IGF-1 and a substance P derivative, FGLM amide (phenylalanine-glycine-leucine-methionine amide), can aid in the healing of persistent corneal epithelial defects in patients with NK [84,175].

The first FDA-approved topical biologic medication in ophthalmology is Cenegermin/Oxervate^®^, a recombinant human NGF protein, specifically for NK [173,176,177], which is fitting as NGF was the first discovered growth factor in 1952 by Rita Levi-Montalcini and Stanley Cohen [178]. Cenegermin resolves roughly 72% of NK cases [179], but its use in other types of ocular perturbations (i.e., ulcerations, keratitis, surface wounds) is not well studied. Further, the use of this drug can present as a challenge to some patients, as it can be costly, difficult to prepare for self-administration, and has an intensive dosing regimen [180,181]. Additionally, there are often issues with patient compliance, as eye drops can cause pain, burning, and blurry vision after use [180].

There are other FDA-approved growth factor-based treatments including Regranex, which is a recombinant PDGF-BB ointment approved for healing diabetic ulcers. However, it is currently not recommended to use on any other types of wounds due to lack of clinical data and testing [182]. There are also some topical recombinant EGF formulations approved in countries outside of the USA for dermal applications [183,184,185] which may be useful for ocular surface wounds.

## 5. Emerging Opportunities

Given the available therapies, why would we need to continue the search? Many of these formulations are made up for a particular subset of patients—like Cenegermin which mainly aids in NK. But, as stated previously, it only aids in about 72% of patients. Further therapies are required to help in preventing corneal blindness due to imbalances in corneal epithelial homeostasis.

### 5.1. EGF: The Catalyst of Corneal Homeostasis Studies

A number of other growth factors have been considered to promote corneal re-epithelialization. Chief among these is topical EGF. Numerous corneal epithelial debridement models show that EGFR activity is both necessary (EGFR inhibitors prevent restoration) and sufficient (treatment with EGF accelerates restoration) for re-epithelialization [58,186]. However, in the clinic, the efficacy of topical EGF on corneal epithelial damage is dependent on the type of the wound. It has shown promise with traumatic ulcers [187,188], surface abrasions/lesions [59], and chemotherapy-induced erosions [151,189]; however, it has limited therapeutic benefit to patients with HSV-derived ulcers, bullous keratopathies, or penetrating keratoplasty [59,190,191].

It has been suggested that the sustained release of EGF is required for optimal EGFR signaling in the corneal epithelium [192]. This is likely due to “receptor desensitization” [193,194] which describes the peak in receptor activity (phosphorylation) following ligand stimulation that attenuates with time. This phenomenon of desensitization has been clearly described both in vitro and in vivo [195,196,197]. Molecular mechanisms include the dephosphorylation of the receptor by phosphatases, internalization of the receptor from the plasma membrane, and post-translational modifications (i.e., ubiquitin) that target the receptor for degradation. Receptor desensitization is a common feature of many receptors that are involved in corneal re-epithelialization and listed in Table 1 [198]. Under normal physiologic conditions, cellular mechanisms like receptor desensitization are critical for maintaining tissue homeostasis and preventing corneal hyperplasia [56,199]. However, under wounded conditions, these mechanisms of receptor inactivation slow the restoration process.

### 5.2. CBL-Mediated Desensitization of RTKs

Many RTKs are negatively regulated by the universal process of receptor ubiquitylation. Ubiquitylation limits receptor signaling, thus it is possible that their contribution to corneal epithelial homeostasis and wound healing is not fully recognized due to desensitization.

Antagonizing receptor ubiquitylation has emerged as a viable method for preventing receptor desensitization, thus sustaining receptor signaling. Ubiquitylation is involved in the endocytic trafficking of activated receptors. Ubiquitylated receptors bind to the endosomal sorting complexes required for transport (ESCRT) machinery on the limiting membrane of the late endosome and internalize into the intraluminal vesicles (ILV) of the late endosome. These late endosomes fuse with the lysosome and transfer the ILVs and its receptor cargo for degradation (Figure 3) [200].

The ubiquitylation proteasome system (UPS) selectively regulates the balance of proteins in the cell. Proteins are conjugated with the 76-amino acid protein ubiquitin (Ub) via a series of enzymatic reactions involving the E1, E2, and E3 Ub ligase machinery that targets proteins for proteasomal or lysosomal degradation [201]. There are two E1 activating enzymes that transfer Ub to one of 38 E2-conjugating enzymes. There are several hundred E3 ligases that mediate the final step of Ub transfer to its specific target. E3 ligases are good pharmacologic targets because only liganded receptors are phosphorylated and have the requisite phosphotyrosine for ubiquitin transfer [45].

Protein ubiquitylation is counter-regulated by deubiquitylating enzymes (DUBs). Several studies highlight the role of E3 ligases in growth factor receptor desensitization. Two of the primary E3 ligases that regulate many RTKs in the corneal epithelium are c-Cbl and Cbl-b. These E3 ligases have roles in other facets of membrane trafficking (i.e., autophagy), which are independent of its E3 ligase activity and have only been observed in cancer cells [202]. The major role of c-Cbl and Cbl-b is downregulation of cell surface receptors.

**Figure 3 cells-12-02730-f003:**
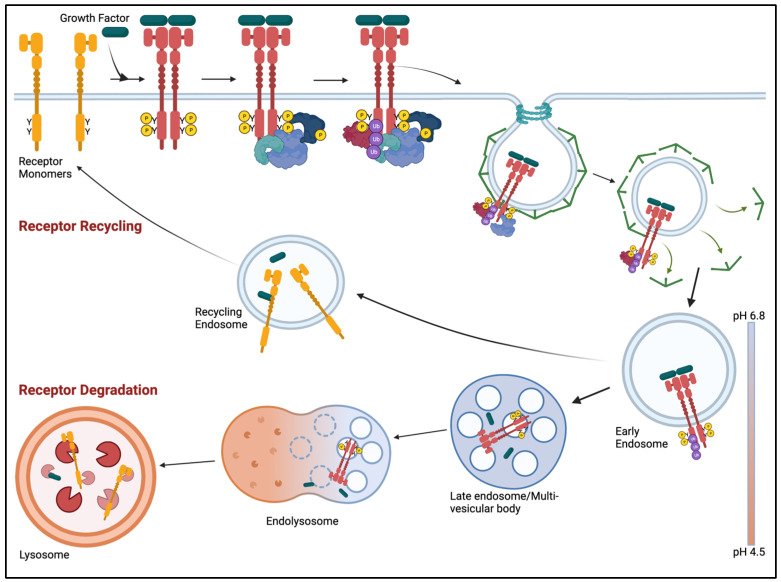
Receptor desensitization. [Starting top left] Ligand induces the dimerization of two receptor monomers, causing conformational changes and leading to auto-transphosphorylation. Phosphorylated tyrosines serve as catalytic or docking sites (See Figure 4). Concomitantly, receptor dimers translocate to clathrin rich membrane domains which invaginate to form a clathrin-coated pit containing the ligand-receptor complex [203]. Dynamin pinches off the membrane and generates a clathrin-coated vesicle [204,205,206]. Clathrin proteins are shed and recycled. The resulting naked vesicle is delivered to and fuses with the early endosome. The early endosome sorts the cargo for its ultimate cellular fate. Receptors can be recycled back to the plasma membrane for additional rounds of activation. Alternatively, receptors can be retained in the early endosome which matures into a late endosome [200]. E3 ubiquitin ligases (i.e., c-Cbl, Cbl-b) bind and transfer ubiquitin to the receptor. Receptor ubiquitylation is a critical modification for endocytosis and allows the receptor to be recognized by the ESCRT complexes. Ubiquitylated receptors bind to ESCRT proteins and become sequestered into intralumenal vesicles (ILV) within the mature late endosome. The late endosome fuses with the lysosome and the cargo is transferred for degradation. Created with BioRender.com (accessed on 17 November 2023).

Evidence to support ubiquitylation as a target comes from studies with the EGFR. The knockdown and/or knockout of c-Cbl decreased EGFR ubiquitylation and increased the rate of corneal epithelial in vitro wound healing [58,65]. Indirect pharmacologic inhibition of ubiquitylation via PP1 (Src inhibitor) resulted in faster in vitro and in vivo corneal re-epithelialization [58]. As indicated in Table 1, multiple receptors are regulated by CBL proteins. Antagonizing CBL activity may be a more universal approach to sustain receptor signaling in the corneal epithelium.

An additional example is the c-Met receptor. For instance, activation of c-Met by HGF promotes corneal epithelial restoration [44,45], but treatment with HGF at supraphysiologic concentrations can limit the therapeutic benefit of the growth factor [207]. Corneal epithelial cells deficient in c-Cbl and Cbl-b demonstrated slowed HGF-driven c-Met trafficking, which resulted in enhanced receptor and effector signaling. The greater magnitude and duration of c-Met phosphorylation in these knockout cells potentiated in vitro wound healing rates 2-fold [45].

### 5.3. HGF: The Multi-Faceted Growth Factor

Although re-epithelialization is the critical first step in corneal wound healing, other aspects such as chronic inflammation and nerve regeneration need to be considered. Additional roles for growth factors receptors include mitogenic [44], anti-fibrotic [47], anti-angiogenic [54], and neurotrophic [208] effects. While most growth factors and their receptors can aid in re-epithelialization, the unique features of the HGF:c-Met signaling axis fulfills many roles within the healing process. 

#### 5.3.1. Inhibition of Fibrosis

Inflammation, in the early stages of healing, is beneficial. Damaged epithelial cells continuously release growth factors like TGF-β and PDGF into the stroma [34,61,209]. When TGF-β and other cytokines are present in high concentrations in the stroma, they differentiate keratocytes into corneal fibroblasts and then to mature myofibroblasts [77,210,211,212,213,214,215]. Mature myofibroblasts are opaque and in chronic wound healing scenarios, they persistently release high levels of disordered extracellular matrix. This clouds the stromal layer of the cornea and impairs vision [77,216,217]. The stromal response to injury will not fully terminate unless the epithelial basement membrane reforms, so the healing of the epithelial layer is crucial for full ocular restoration.

The balance between anti-inflammatory (i.e., HGF and EGF) and pro-inflammatory (TGF-β) signaling mediators determine the extent of tissue damage. TGF-β levels rise as injuries become chronic, leading to fibrosis [218,219]. HGF can inhibit TGF-β production by upregulating Smad7 [47,220,221], which in turn prevents myofibroblasts from maturing [77,222,223]. Activation of the EGFR can promote nuclear factor κβ (NF-κβ) activity, which inhibits TGF-β signaling [224]. Further, HGF can suppress pro-inflammatory cytokines IL-1, IL-6, and IL-18 that are released from macrophages [225]. Lastly, combination gene therapy with HGF and bone morphogenic protein 7 (BMP7) decreased corneal fibrosis following in vivo rabbit corneal alkali burn [226,227].

**Figure 4 cells-12-02730-f004:**
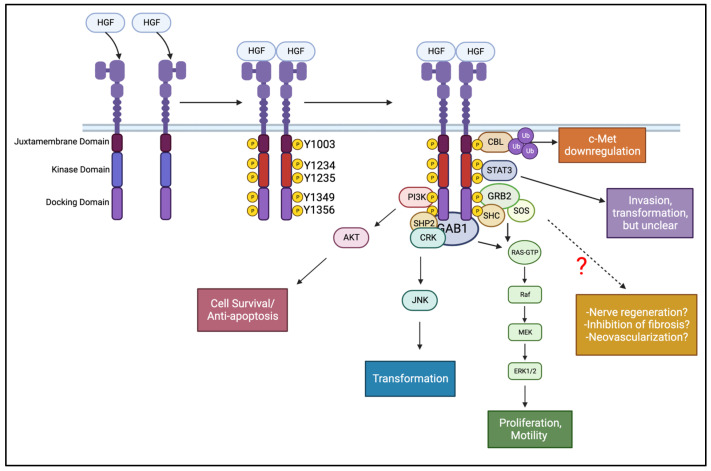
Example of the c-Met receptor’s activation pathways. Receptor activation begins with two HGF molecules binding to two c-Met monomers, promoting the dimerization and stabilization of the complex [228]. Dimerization of the transmembrane monomers allows for the kinase domains on each receptor to activate via auto-transphosphorylation. Once the catalytic domains are active (Y1234/Y1235), they mediate the phosphorylation of tyrosine residues in both the juxtamembrane (Y1003) and docking domains (Y1349/Y1356). The phosphorylated docking domain allows for scaffold (i.e., GAB1, Grb2) and effector proteins (i.e., STAT3, PI3K) to bind. Effector proteins can be phosphorylated directly by the receptor, or by binding onto scaffold proteins to be brought in range of the c-Met catalytic domain. Different outcomes are driven by which effector proteins are activated. A few pathways are clearer than others, namely the Ras/Raf/Mek/ERK1/2 pathway and its involvement in proliferation and cell motility [229]. It is also well established that PI3-K works through Akt to prevent apoptosis in corneal epithelial cells [230]. Other pathways, including the CRK-JNK for transformation [231,232] and STAT3 for invasion [233,234], have various results, so the outcome of signaling through these pathways may be determined by cell type. One gap in the research is what pathways are activated and how c-Met plays a role in nerve regeneration, preventing fibrosis, and angiogenesis. Following the phosphorylation of Y1003 in the juxtamembrane domain, E3 ubiquitin ligases like c-Cbl or Cbl-b can bind and transfer ubiquitin molecules to the receptor, ultimately ending with receptor degradation (see Figure 4). Created with BioRender.com (accessed on 17 November 2023).

#### 5.3.2. Corneal Neovascularization

Another clinical complication in corneal wound healing is neo-vascularization. The absence of blood and lymphatic vessels in the cornea keeps it transparent and allows light to pass through and refract on the retina. Under normal conditions, the protein Notch1 suppresses VEGF expression in the cornea. When Notch1 activity decreases or when VEGFR is hyperactivated, neovascularization branches from the blood vessels of the limbal vascular plexus [235]. The downregulation of TGF-β in an in vivo burn model decreases not only infiltrating inflammatory cells and disordered ECM, but also the formation of new vessels [236].

HGF has been implicated as a regulator of angiogenesis, but there is no unifying model. In some tissues, HGF is a potent angiogenic factor, particularly in the retina [237,238,239]. For the anterior eye, some studies indicate that the inhibition of HGF via siRNA prevents VEGF-dependent corneal neovascularization, but this was accompanied by decreased epithelial proliferation and increased incidence of apoptosis [240]. Conversely, implantation of pellets into the mouse cornea that contain an HGF derivative, H-RN, *also* prevented VEGF-driven angiogenesis [52,55]. Together, these findings suggest a role for HGF in regulating corneal neovascularization, but further investigation is needed.

#### 5.3.3. Neuro-Regeneration

Restoring the corneal epithelium provides a critical barrier to foreign agents and limits the chances of infection. However, without neuronal restoration, the eye lacks the necessary sensitivity it needs to avoid recurrent erosions and surface shedding.

c-Met has roles in in vivo [49], ex vivo [241], and in vitro [51,242] neuronal growth models. In aging mice, decreases in c-Met activation parallel the loss of nerve regeneration [50]. c-Met has also been implicated in the formation of new synapses [48]. However, there is no literature surrounding c-Met and HGF involvement in intraepithelial corneal nerve communication and wound healing, which is a gap in the field that needs to be addressed.

## 6. Conclusions

Damage to the corneal epithelium is a component of almost all corneal injuries. In addition to intense pain, perturbation to the epithelial layer makes the eye susceptible to infection and potentially loss of vision. Growth factor receptors have a central role in corneal epithelial homeostasis and regeneration, however, due to our incomplete understanding of their signaling, they have not yet reached their full potential as a therapy. Critical next steps include identifying the most efficacious mediators of corneal regeneration and, perhaps most importantly, identifying limitations to their use. Uncovering the fundamental mechanisms of how these proteins work will serve as a foundation for developing new therapies to treat the millions of individuals affected each year by corneal perturbations.

## Figures and Tables

**Figure 1 cells-12-02730-f001:**
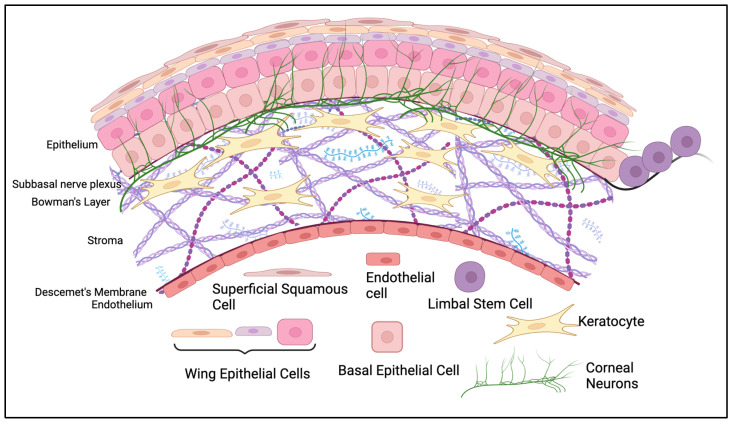
Anatomical structure of the corneal epithelium. The most anterior epithelial layer is made of 5–7 layers of epithelial cells. Basal epithelial cells arise from limbal stem cells (LSCs) that move centripetally into the cornea from the limbus. As basal epithelial cells move through their life cycle, they become smaller, move anteriorly, and are eventually shed as superficial squamous cells. The epithelium lies above Bowman’s Layer and the subbasal plexus, which are anterior to the stroma. The stroma is the thickest layer and is mainly populated by keratocytes, which release extracellular matrix and collagen to maintain the transparency of the cornea. The stroma is separated from the most posterior layer of the cornea, the endothelium, by Descemet’s membrane. The endothelium is made up of a single layer of endothelial cells that tightly regulate fluid dynamics from the aqueous humor. Created with BioRender.com (accessed on 17 November 2023).

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
