# Peer review of "Modulating Growth Factor Receptor Signaling to Promote Corneal Epithelial Homeostasis"

_cells, 2023, doi:10.3390/cells12232730_

Round 1

Reviewer 1 Report

Comments and Suggestions for Authors

Summary:

This review entitled Growth Factor Receptor Signaling in the Corneal Epithelium provides a useful description of the different roles of growth factors and their cognate receptors in mediating control of corneal renewal under normal conditions and in different wound injury models. It can help guide efforts to identify potential drug targets that may improve therapeutic management of corneal disease and injury in a clinical setting.   Despite its very extensive literature review, the authors overlooked describing some other very pertinent findings that are of particular interest. These omissions are listed below.

Concerns:

1)      There is no mention of studies describing how epidermal growth factor receptor (EGFR) activation can be modified to alter downstream signaling events modulating corneal epithelial renewal.  Specifically, it is recommended that they think about citing references such as PMID 19443725.

 2)      They need to cite at least one study showing the involvement of an endogenous EGFR ligand and the mechanism by which it is generated in response to wounding.

3)      It is perplexing that there is no mention of the numerous roles of different transient receptor potential (TRP) channel isoforms in modulating EGFR control of epithelial wound healing in different wound healing models. For example, studies are overlooked describing context differences in the effects of TRPV1 on wound healing. Specifically, TRPV1 activation induces scarification, inflammation and neovascularization in an alkali burn model  whereas it accelerates renewal in an incision wound model. At the very least, it will be instructive to cite a 2022 review describing the complex roles of TRP channel activation in modulating corneal function under different environmental conditions (PMID: 35058918). In the last decade, there are about 12 different reviews that the authors can select from for their description of the multiple roles of TRP function in the cornea.

4)      Earlier citations are warranted describing the important growth factor role of EGF in the cornea. 

 5)      Grammatical error: Line 238 Change is to it.

Comments on the Quality of English Language

English is acceptable.

Author Response

We thank the reviewers for their close reading and comments on our review article. The suggested changes have been implemented and we believe make the review article stronger. Please see below how we executed the revisions. 

Reviewer 1- Revisions highlighted in YELLOW

1) There is no mention of studies describing how epidermal growth factor receptor (EGFR) activation can be modified to alter downstream signaling events modulating corneal epithelial renewal.  Specifically, it is recommended that they think about citing references such as PMID 19443725.

  • Discussion of how effectors can modulate receptor signaling has been added into lines ­­­­­­­­­­­­191-193.

 2) They need to cite at least one study showing the involvement of an endogenous EGFR ligand and the mechanism by which it is generated in response to wounding.

  • The following citations were added into the manuscript in line 145.
    • Breyer JA, Cohen S. The epidermal growth factor precursor isolated from murine kidney membranes. Chemical characterization and biological properties. J Biol Chem. 1990 Sep 25;265(27):16564-70. PMID: 2398064.
    • Ke-Ping Xu, Yu Ding, Jianhua Ling, Zheng Dong, Fu-Shin X. Yu; Wound-Induced HB-EGF Ectodomain Shedding and EGFR Activation in Corneal Epithelial Cells. Invest. Ophthalmol. Vis. Sci. 2004;45(3):813-820. https://doi.org/10.1167/iovs.03-0851.

3) It is perplexing that there is no mention of the numerous roles of different transient receptor potential (TRP) channel isoforms in modulating EGFR control of epithelial wound healing in different wound healing models. For example, studies are overlooked describing context differences in the effects of TRPV1 on wound healing. Specifically, TRPV1 activation induces scarification, inflammation and neovascularization in an alkali burn model, whereas it accelerates renewal in an incision wound model. At the very least, it will be instructive to cite a 2022 review describing the complex roles of TRP channel activation in modulating corneal function under different environmental conditions (PMID: 35058918). In the last decade, there are about 12 different reviews that the authors can select from for their description of the multiple roles of TRP function in the cornea.

  • Discussion of TRP channels and how they can transactivate growth factor receptors has been added in lines 195-209.

4) Earlier citations are warranted describing the important growth factor role of EGF in the cornea. 

  • The authors added more citations from earlier studies showing the importance of EGF in corneal wound healing to Table 1:
    • Daniele S, Frati L, Fiore C, Santoni G. The effect of the epidermal growth factor (EGF) on the corneal epithelium in humans. Albrecht Von Graefes Arch Klin Exp Ophthalmol. 1979 May 7;210(3):159-65. doi: 10.1007/BF00414564. PMID: 315726.
    • Savage CR Jr, Cohen S. Proliferation of corneal epithelium induced by epidermal growth factor. Exp Eye Res. 1973 Mar;15(3):361-6. doi: 10.1016/0014-4835(73)90151-6. PMID: 4695441.
    • Tao W, Liou GI, Wu X, Abney TO, Reinach PS. ETB and epidermal growth factor receptor stimulation of wound closure in bovine corneal epithelial cells. Invest Ophthalmol Vis Sci. 1995 Dec;36(13):2614-22. Erratum in: Invest Ophthalmol Vis Sci 1996 Sep;37(10):1937. PMID: 7499084.

5) Grammatical error: Line 238 Change is to it.

  • Made this change. Thank you to the reviewer for catching it! (Now line 275).

Reviewer 2 Report

Comments and Suggestions for Authors

It's an interesting review.

  1. Line88-94. It mentioned tear dysfunction in Sjogren syndrome. How about in dry eye syndrome? Is there any change in growth factor in tear?
  2. Line 110-118. Recently many evidence suggest that there are non-myelinating Schwann cells in cornea.
  3. Line 264-274. Receptor desensitization occurs in some types of wound and results in delayed wound healing even though with topical treatment of growth factors. This is very interesting topic. It may make this review more interesting if the authors can summarize and give more details about what conditions have receptor desensitization and which growth factors are affected.
  4. Line 281-309. The authors summarized how receptor proteins are degraded by endocytosis and UPS. However, in addition to endocytosis and UPS, autophagy is another major pathway by which proteins can be degraded. The authors should also mention autophagy here.

Author Response

We thank the reviewers for their close reading and comments on our review article. The suggested changes have been implemented and we believe make the review article stronger. Please see below how we executed the revisions.

Reviewer 2- Revisions highlighted in BLUE

1) Line 88-94. It mentioned tear dysfunction in Sjogren syndrome. How about in dry eye syndrome? Is there any change in growth factor in tear?

  • Another reviewer also asked for more discussion of disease states. We added in discussion in lines 211-237.

2) Line 110-118. Recently many evidence suggest that there are non-myelinating Schwann cells in cornea.

  • An additional sentence was added in about non-myelinating Schwann cells present in the cornea in lines 100-101.

3) Line 264-274. Receptor desensitization occurs in some types of wound and results in delayed wound healing even though with topical treatment of growth factors. This is very interesting topic. It may make this review more interesting if the authors can summarize and give more details about what conditions have receptor desensitization and which growth factors are affected.

  • The authors edited the table and added in a line discussing that most of the growth factors listed in the table are affected by desensitization (lines 336-339). There aren’t necessarily differences in desensitization between wound types, it is a process that occurs normally in both wounded and unwounded conditions.

4) Line 281-309. The authors summarized how receptor proteins are degraded by endocytosis and UPS. However, in addition to endocytosis and UPS, autophagy is another major pathway by which proteins can be degraded. The authors should also mention autophagy here.

  • Discussion of autophagy has been added in lines 367-369.

Reviewer 3 Report

Comments and Suggestions for Authors

The review deals with growth factors and their receptors in the corneal epithelium. The abstract already gives the impression that tear growth factors are important for the maintenance of the corneal epithelium, yet the epithelial cells produce their own factors, which needs to be mentioned along the lines of Figure 1. The review is a little disorganized, with undue emphasis on HGF/c-met and poor description of other GF/GFR systems. The title does not really reflect the content as signaling is not discussed in detail. Other concerns are listed below.

1. “Intraepithelial corneal nerves (ICNs)”. Hopefully, the authors mean “Intraepithelial corneal nerve endings”.

2. When talking about existing drugs, the authors may need to mention topical formulations other than Cenegermin. Such drugs may be used off label for eye conditions, e.g., Regranex/PDGF approved for diabetic foot ulcers. Topical recombinant EGF is used in clinical dermatology and could also be beneficial for ophthalmic diseases. Some approved formulations exist in several countries (Regen-D™ 150, India; Heberprot-P® 25/75, Cuba; Easyef, South Korea).

3. Table 1 lists epithelium-related growth factors studied largely in the normal corneas. It would be important to include studies on disease-altered growth factors.

4. Figure 3 only deals with c-met signaling, whereas other growth factor receptors are left out. The authors may need to rethink this figure to add other receptors or remove it. Figure 4 is again about c-met. First, it seems to be out of context, and second, what about other growth factors and their receptors?

5. In section 4.1, the description of amniotic membrane seems to be out of place. Despite the presence of some growth factors in it, their role in AM effects has not been proven or well tested.

6. The role of growth factors in corneal wound healing has not been covered well, although pertinent detailed reviews exist. Please expand.

Author Response

Reviewer 3- Revisions highlighted in GREEN

We thank the reviewers for their close reading and comments on our review article. The suggested changes have been implemented and we believe make the review article stronger. Please see below how we executed the revisions. 

The review deals with growth factors and their receptors in the corneal epithelium. The abstract already gives the impression that tear growth factors are important for the maintenance of the corneal epithelium, yet the epithelial cells produce their own factors, which needs to be mentioned along the lines of Figure 1. The review is a little disorganized, with undue emphasis on HGF/c-met and poor description of other GF/GFR systems. The title does not really reflect the content as signaling is not discussed in detail. Other concerns are listed below.

  • Thank you to the reviewer for their input. We have changed the title of the manuscript and, in our opinion, have organized it in a much better fashion. We also updated Table 1 to really reflect what we are trying to communicate. We have also made sure to emphasize that epithelial cells produce their own factors.
  1. “Intraepithelial corneal nerves (ICNs)”. Hopefully, the authors mean “Intraepithelial corneal nerve endings”.
    • Thank you to the reviewer for catching this mistake! This has been changed in line 89.
  1. When talking about existing drugs, the authors may need to mention topical formulations other than Cenegermin. Such drugs may be used off label for eye conditions, e.g., Regranex/PDGF approved for diabetic foot ulcers. Topical recombinant EGF is used in clinical dermatology and could also be beneficial for ophthalmic diseases. Some approved formulations exist in several countries (Regen-D™ 150, India; Heberprot-P® 25/75, Cuba; Easyef, South Korea).
    • Discussion of these recombinant treatments have been added in our new section 4.2.2, lines 305-310.
  1. Table 1 lists epithelium-related growth factors studied largely in the normal corneas. It would be important to include studies on disease-altered growth factors.
    • Another reviewer also asked for more discussion of disease states. We added in discussion in our new section 3.4, lines 211-237.
  1. Figure 3 only deals with c-met signaling, whereas other growth factor receptors are left out. The authors may need to rethink this figure to add other receptors or remove it. Figure 4 is again about c-met. First, it seems to be out of context, and second, what about other growth factors and their receptors?
    • To address this, we have better organized the manuscript. First, figure 3 and 4 have been switched in order, and discussion of c-Met has been held until the end of the manuscript.
    • The new figure 3 (Line 370) is more generic of growth factor receptors instead of just being c-Met.
    • As for the other growth factors, we wanted to shine a light on the c-Met signaling axis. We included the other growth factors in Table 1.
  1. In section 4.1, the description of amniotic membrane seems to be out of place. Despite the presence of some growth factors in it, their role in AM effects has not been proven or well tested.
    • A disclaimer clause has been added in lines 279-280.
  1. The role of growth factors in corneal wound healing has not been covered well, although pertinent detailed reviews exist. Please expand.
    • Upon reflecting on all three reviewers' comments, we have updated the manuscript title and organization. These changes better reflect our major discussion points: 1) growth factor receptor signaling is an integral part of corneal epithelial homeostasis, 2) blocking growth factor desensitization is a viable strategy for enhancing signaling by the many growth factor receptors that are subject to desensitization, and 3) to fully restore the corneal epithelium we should consider growth factors to do more than promote the epithelial layer. We have placed more emphasis on our second and third points and provide references to material that has already been reviewed and could not be covered in the scope of this manuscript.  
    • Additional review articles about growth factor receptors in corneal wound healing have been added into line 113.

Round 2

Reviewer 1 Report

Comments and Suggestions for Authors

The authors have adequately addressed all the previous concerns.  There are no other concerns requiring their attention. 

Reviewer 3 Report

Comments and Suggestions for Authors

The comments have been addressed well.